# Accuracy and Speed Improvement of Event Camera Motion Estimation Using a Bird’s-Eye View Transformation

**DOI:** 10.3390/s22030773

**Published:** 2022-01-20

**Authors:** Takehiro Ozawa, Yusuke Sekikawa, Hideo Saito

**Affiliations:** 1Graduate School of Science and Technology, Keio University, Yokohama 223-8522, Japan; hs@keio.jp; 2Denso IT Laboratory, Inc., Tokyo 150-0002, Japan; ysekikawa@mail.d-itlab.co.jp

**Keywords:** motion estimation, event-based camera, bird’s-eye view, autonomous vehicles

## Abstract

Event cameras are bio-inspired sensors that have a high dynamic range and temporal resolution. This property enables motion estimation from textures with repeating patterns, which is difficult to achieve with RGB cameras. Therefore, motion estimation of an event camera is expected to be applied to vehicle position estimation. An existing method, called contrast maximization, is one of the methods that can be used for event camera motion estimation by capturing road surfaces. However, contrast maximization tends to fall into a local solution when estimating three-dimensional motion, which makes correct estimation difficult. To solve this problem, we propose a method for motion estimation by optimizing contrast in the bird’s-eye view space. Instead of performing three-dimensional motion estimation, we reduced the dimensionality to two-dimensional motion estimation by transforming the event data to a bird’s-eye view using homography calculated from the event camera position. This transformation mitigates the problem of the loss function becoming non-convex, which occurs in conventional methods. As a quantitative experiment, we created event data by using a car simulator and evaluated our motion estimation method, showing an improvement in accuracy and speed. In addition, we conducted estimation from real event data and evaluated the results qualitatively, showing an improvement in accuracy.

## 1. Introduction

### 1.1. Backgrounds

Camera motion estimation is one of the most important technologies in many applications, such as automatic driving and assistive technologies. In particular, motion estimation by pointing a camera at a road surface has an advantage in that objects that can affect it, such as automobiles and pedestrians, are usually not easily visible [1,2,3,4,5,6]. Teshima et al. [1] proposed a method to estimate the position and orientation of a vehicle from sequential images of the ground using the known homography obtained from a normal camera. Saurer et al. [2] proposed a method to find different minimal solutions for egomotion estimation of a camera based on homography knowing the gravity vector between images. Gilles et al. [3] proposed a method based on unsupervised deep learning for motion estimation with a downward-looking camera. However, in spite of active research, camera motion estimation by using the road surface has the following problems: it is difficult to extract and match feature points robustly from noisy ground textures; the high-speed movement of cameras causes motion blur; it often involves challenging illumination conditions.

Event cameras [7] have been developed with features that can address these problems. Instead of recording an image intensity at a synchronous frame rate, event cameras record asynchronous positive and negative spikes, called events, in response to changes in the brightness of a scene. Therefore, the output is not an intensity image, but a stream of asynchronous events, where each event consists of its space–time coordinates. The advantages of event cameras include their high temporal resolution (μs order), high dynamic range (130 dB), and low power consumption [8,9], as well as the ability to acquire data without blurring, even when the event camera itself is moving at high speed. However, they are fundamentally different in their operating characteristics from traditional frame-based cameras; thus, they require completely new algorithms.

As one of the methods for motion estimation using event cameras, Gallego et al. [10,11] and Stoffregen et al. [12] showed that the spatio-temporal trajectory of an event when capturing a plane can be modeled by a homographic transformation, and the motion can be determined by maximizing the contrast of an image of a warped event (IWE) along the trajectory (i.e., contrast maximization).

### 1.2. Motivations

As shown by Nunes et al. [13], contrast maximization has the problem that the loss function may have multiple extrema. This implies that the estimation results are highly dependent on the initial values. In addition, especially when all the events are warped on the same point or line, the contrast is larger, and the entropy is smaller (as compared with when they are warped with the correct value); see Figure 1. Therefore, depending on the initial values, it is possible to estimate parameters that differ significantly from the correct values. This problem can occur when estimating motion in three-dimensional Euclidean space, such as in homography. Hence, it is necessary to solve this problem when performing camera motion estimation using an event camera that is diagonally facing the ground.

### 1.3. Objectives

Our research objectives were to solve the problem of event camera motion estimation using contrast maximization, where the loss function has multiple extrema, and to show its effectiveness experimentally. To solve this problem, we propose a method for motion estimation by maximizing the contrast in the bird’s-eye view space. Figure 2 shows the overview of our method. Assuming that the event camera is mounted on a vehicle and its mounting position is known, homographic information can be obtained in advance. Furthermore, assuming that the vehicle moves parallel to the road, the motion of the event camera can be modeled in 3DoF. Using the obtained homography, a bird’s-eye view transformation can reduce the 3D motion estimation problem to the 2D planar motion estimation problem. We empirically found that this transformation makes the loss function convex around the true value and improves the accuracy and speed of the estimation.

The proposed method was quantitatively evaluated using event sequences generated using the CARLA simulator [14]. We compared the accuracy and speed of the motion estimation results of the proposed method and the existing methods and show that the proposed method was better in both cases. In the above experiment, we also confirmed that the loss function was convex around the true value by performing a bird’s-eye view transformation. We also qualitatively evaluated the proposed method by estimating the motion of an actual event camera installed on a dolly.

### 1.4. Related Works

As it is a similar problem to camera motion estimation by using the road surface, optical flow estimation with event cameras can be mentioned. In early research, a method using intensity image reconstruction by an accumulation of events was proposed [15]. Furthermore, several methods specific to event cameras using local plane fitting in space–time were proposed [16,17,18,19,20]. Almatrafi et al. [21] introduced the concept of the distance surface and used it for a successful estimation of optical flow with high accuracy. Recently, methods for optical flow estimation using machine learning have been successfully applied [22,23,24,25,26,27].

Motion estimation using event cameras has also been studied. Weikersdorfer et al. [28,29] captured images of a ceiling using an event camera mounted upward and used particle filters to estimate the position of the camera, which was moving parallel to the plane. However, this method requires capturing a plane in parallel and a black-and-white line pattern in the scene. Kim et al. [30,31] proposed a method to jointly estimate six degrees of freedom (DoFs) of the camera, depth, and intensity gradient of the scene. The results were limited to small-scale environments and motions. Gallego et al. [32] proposed to estimate rotational motion by accumulating events over a small time window into an image and maximizing the variance of the image. Gallego et al. [10,11] proposed a contrast maximization framework, based on the method of [32]. This framework maximizes the contrast of the image by warping events according to the model parameters and accumulating the warped events onto the IWE. Nunes et al. [13] proposed a method called entropy minimization, based on contrast maximization, that performs motion estimation without projecting onto the IWE. However, as shown by Nunes et al. [13], both contrast maximization and entropy minimization have the problem that the loss function may have multiple extrema.

In response to this problem, Liu et al. [33] proposed a method for finding the global maximum solution of a rotational motion using a branch and bound (BnB) approach for contrast maximization. Peng et al. [34] also proposed a method for finding the global maximum solution using BnB and achieved planar motion estimation from an event camera mounted downward on an automatic guided vehicle (AGV). However, when mounting an event camera on a car, it is difficult to capture the road surface vertically.

### 1.5. Contributions

The main contributions of this paper are as follows:We empirically found that the loss function of contrast maximization becomes convex around the true value by performing a bird’s-eye transformation to the event data. In doing this, we accomplished the following:
-The initial value of the estimation can be taken widely;-The global extremum will be the correct value.The effectiveness of the proposed method was demonstrated using both synthetic and real data.

Note that the proposed method is based on empirical results and lacks mathematical analysis. However, the effectiveness of our method was clearly demonstrated through experiments.

The rest of this paper is organized as follows: Section 2 describes our proposed method. In Section 3, we discuss the results of quantitative experiments conducted with event data generated by the CARLA simulator and qualitative experiments conducted with real event data. Section 4 presents a discussion of the results, the limitations, and the future works. Finally, Section 5 summarizes the overall results of this study.

## 2. Materials and Methods

### 2.1. Event Data Representation

An event camera asynchronously outputs the luminance change of each pixel, called an event. Suppose there exists a given set of events $E=e_{k}$ in the time window [0,T], where $e_{k}=\left(x_{k},t_{k},p_{k}\right)$. Here, $x_{k}$ is the coordinate where the event triggered, $t_{k}$, is the timestamp where the event occurs with a precision of microseconds and $p_{k}\in \left\{-1,+1\right\}$ is the polarity of the event, which indicates brighter or darker changes when the logarithmic intensity changes above a threshold at a certain pixel x=(x,y).

### 2.2. Contrast Maximization

Gallego et al. [10] proposed a unified framework called contrast maximization, which can be applied to various problems in event-based vision. Because the proposed method relies on contrast maximization, it is briefly reviewed here.

Events originate from edges with high luminance gradients, as indicated by their properties. The goal of contrast maximization is finding the trajectory of events that originate from the same edge in XYT space caused by the camera or scene motion. Given their coordinates and timestamps, events originating from the same edge can be aggregated to a single point by warping along the trajectory. The warped events projected onto the XY image plane are called the image of warped events (IWEs). Once the correct trajectory is found, the IWE becomes a sharp image as an edge map. Gallego et al. [10] estimated the trajectory so that the contrast of the IWE is maximized.

Given *N* events, $E={\left\{e_{k}\right\}}_{k=1}^{N}$, let $x_{k}^{\prime }=W\left(x_{k}^{\prime },t_{k};\theta \right)$ be the function that warps $e_{k}$ along the trajectory to $x_{k}^{\prime }$. θ is the parameter of the warping, and the IWE is obtained by accumulating the events warped by W as follows:(1)$$I\left(x;\theta \right)≐\sum_{k=1}^{N}\delta \left(x-W(x_{k}^{\prime },t_{k};\theta )\right),$$
where δ(·) is a delta function, approximated by a Gaussian. Now, the variance of the IWE is defined as follows:(2)$$\mathrm{Var}\left(I;\theta \right)=\frac{1}{N}\sum_{x}{\left(I\left(x;\theta \right)-\mu \left(I;\theta \right)\right)}^{2}$$
(3)$$\mu \left(I;\theta \right)=\frac{1}{N}\sum_{x}I\left(x;\theta \right).$$

The goal of contrast maximization is to find the $\theta^{*}$ that maximizes the variance of the IWE,
(4)$$\theta^{*}=\arg\max_{\theta }\mathrm{Var}\left(I;\theta \right).$$

Even surfaces with few robust features, such as many roads, can be tracked using the contrast maximization described above.

Camera motion estimation in planar scenes such as a road surface can be achieved by using the homography model. This model has eight degrees of freedom (DoFs) and can be parameterized by the 3D angular velocity, $w=(w_{x},w_{y},w_{z})$; the 3D velocity, $v=(v_{x},v_{y},v_{z})$; and the normalized normal vector of the target plane, $n=(n_{x},n_{y},n_{z})$, expressed as:(5)$$W\left(x_{k},t_{k};\theta \right)\propto H^{-1}\left(t_{k};\theta \right)\left(\begin{array}{c} x_{k}\\ 1 \end{array}\right),$$
where $\theta ={\left(w^{⊤},v^{⊤},n^{⊤}\right)}^{⊤}$ are the model parameters and H is the homography matrix.

### 2.3. Bird’s-Eye View Transformation

Figure 3 shows the flow of our method. Generally, a vehicle moves parallel to a flat road surface without changing its height. Therefore, the motion of the onboard event camera is limited to a plane parallel to the road surface because the camera also moves parallel to the road surface. Therefore, we propose a method to reduce the number of estimation parameters by performing a bird’s-eye view transformation on the event data to convert them to the viewpoint of a virtual camera pointed perpendicular to the road surface, resulting in a two-dimensional problem of motion estimation.

A bird’s-eye view conversion of images from conventional cameras (with the concept of frames) requires interpolation processing. However, since the pixels of an event camera react individually to changes in intensity, the event data are sparse, unlike a conventional image. Therefore, it can be converted more efficiently.

Suppose that the installation angle of the camera is known when the event camera is mounted on the vehicle; the homography for the bird’s-eye view transformation can then be calculated as follows:(6)$$H_{ab}=K_{b}\cdot \left(R_{ab}-t_{ab}n^{T}\right)\cdot K_{a}^{-1},$$
where *a* is the event camera and *b* is a virtual camera oriented perpendicular to the road surface. $R_{ab}$ and $t_{ab}$ are the rotation matrix and translation vector between *a* and *b*, respectively. n is the normal vector to the road surface; $K_{a}$ and $K_{b}$ are the internal parameters of each camera. Let ϕ be the angle between the normal of the road surface and the optical axis of the event camera, which is obtained as $n={\left[0,-\sin\phi ,-\cos\phi \right]}^{⊤}$.

With this homography, $H_{ab}$, we can transform the event as follows:(7)$${\hat{x}}_{a}=H_{ab}x_{b}.$$

The events transformed by (Equation 7) are converted to the viewpoint of a virtual camera moving parallel to the road surface. With this transformation, the motion estimation of the event camera is reduced to a two-dimensional translation and rotation problem. This model has three DoFs and can be parameterized by the angular velocity, ω, and the 2D velocity, $v=(v_{x},v_{y})$, expressed as:(8)$$W\left(x_{k},t_{k};\theta \right)=\left(\begin{array}{cc} R^{-1} & t\\ 0^{T} & 1 \end{array}\right)\left(\begin{array}{c} x_{k}\\ 1 \end{array}\right),$$
where $\theta =(\omega ,v_{x},v_{y})$ are the model parameters shared by all the pixels, $R=\exp(\Delta t_{k}\omega )$ is the 2 × 2 rotation matrix where exp is the exponential map, and $t=\Delta t_{k}v$ is the translation 2D vector.

## 3. Results

The novelty of this method is that we empirically found that the bird’s-eye view transformation makes the loss function convex around the true value, which leads to fast and accurate motion estimation. To demonstrate this, we performed the following three experiments using both synthetic and real datasets.

In the first experiment, to demonstrate the improvement in accuracy and speed of our method, we generated event data acquired from a car driving on a road using the CARLA simulator [14] and compared the accuracy and speed of the proposed method and existing method. In the second experiment, we compared the loss landscape near the ground truth using the same data as in the first experiment and showed that the non-convex loss function becomes convex by performing a bird’s-eye view transformation. In the third experiment, to demonstrate the applicability of our method, we applied our method to real data captured by an event camera and evaluated the results. All experiments were conducted on a system consisting of an Intel Core i7-6800K CPU and 64 GB of system memory, using Python for the implementation.

### 3.1. Accuracy and Speed Evaluation

In this section, we describe and report on the evaluation of the accuracy and speed improvement provided by the bird’s-eye view transformation of the event data. In order to compare the proposed method with bird’s-eye view transformation and the existing method without bird’s-eye view transformation, we conducted an experiment to estimate the parameters on the data generated by CARLA, which simulates a vehicle-mounted event camera (see Figure 4). The event camera was placed at a height of 1 m from the ground at an angle of 45${}^{\circ }$, and the event generation algorithm implemented in the CARLA simulator was used to create the data (similar to ESIM [35]).

For parameter estimation, we used the entropy minimization method [13] as a comparison, whose source code is publicly available. This method defines entropy instead of contrast and minimizes it to estimate the parameters. For the created data, we performed homographic motion estimation using the entropy minimization method and isometric motion estimation using the bird’s-eye view transformation and then compared the speeds and accuracies of the estimations. The motion to be estimated is the angular velocity (*w*) and velocities ($v_{x},v_{y}$) of the event camera.

For the accuracy comparison experiments, we prepared three types of initial values for estimation: (a) using the previous estimation result, (b) using zero, and (c) using the previous estimation result, but with the exception of using zero if the distance from the true value was greater than the threshold.

The results of the accuracy comparison experiment are shown in Figure 5. As can be seen from the figure, the homographic motion estimation in Condition (a) estimated a value that was far from the ground truth. Furthermore, because the previous estimation result was used as the initial value of the next estimation, the estimates continued on incorrectly. Figure 6 shows the events after the warp in the XYT space, and we can see that there was a problem of events being warped to a straight line, as described in Section 1. This problem occurs because when events are warped on a straight line, the entropy is smaller than when it is warped on a correct value. This problem can occur when estimating velocities in three dimensions. In the proposed method, this problem is avoided by reducing the problem to a 2D velocity estimation by performing the bird’s-eye view transformation; see Figure 7. In Condition (b), each time we used the initial value of zero, which was close to the true value, the aforementioned problem was avoided. However, it can be seen that even in this case, the proposed method successfully improved the accuracy. This can be attributed to the fact that the loss function had multiple extrema around the true value and thus fell into a local solution when performing homographic motion estimation, as shown in the following experiment. In the proposed method, the bird’s-eye view transformation is used to make the loss function convex around the true value, which enables relatively accurate estimation. In Condition (c), it was shown that both methods could estimate relatively accurately by using the ground truth information. This means that if the initial values can be set appropriately, homographic motion estimation can be relatively accurate, but it is difficult to set the initial values properly in practice.

For the computational complexity comparison experiments, we extracted four scenes from our dataset (see Figure 8) and measured the number of iterations and computation time until the estimation was completed.

The results of the computational complexity comparison experiment are shown in Table 1. As can be seen from the table, the bird’s-eye view transformation succeeded in reducing the computational complexity, both in terms of the number of iterations and the time required for each iteration. This indicates that reducing the estimated parameters by the bird’s-eye view transformation not only reduced the computational complexity of each iteration, but also improved the ease of convergence.

### 3.2. Loss Landscape

In this section, we show experimentally that the bird’s-eye view transformation makes the loss function in the motion estimation convex around the true value. We extracted 10,000 events at t=270s from the data used in Section 3.1 and compared the distribution of the entropy when warping by homography and when warping by isometry with a bird’s-eye view transformation. To compare the loss landscape, we computed the entropy around the ground truth and visualized using Matplotlib libraries [36].

Figure 9 shows the distribution of the entropy near the true value with and without the bird’s-eye view transformation. In the case of homographic motion estimation, we can see that several local extrema appeared near the true value. In addition, over a wider range, local extrema with lower entropy than the true value appeared. In contrast, in the case of the isometric motion estimation with a bird’s-eye view transformation, the distribution of the entropy was convex over a somewhat wide range. Therefore, it can be said that the bird’s-eye view transformation helped with the relatively fast convergence of the correct solution value, while preventing falling into a local solution.

### 3.3. Evaluation with Real Data

In this section, we describe the application of our method to real data captured by the event camera and the evaluation of the results in order to show the potential of our method for real-world applications. As shown in Figure 10, an event camera was mounted on a dolly angled (non-parallel) to the surface of interest, and the event camera motion estimation was performed when running on a carpeted surface. The event camera used was a Prophesee Gen3 VGA (and its supporting evaluation kit, EVK), installed at an angle of 45${}^{\circ }$ to the ground. As a reference value, motion estimation with the camera facing the front was also performed at the same time using ARKit4 (iOS). In this experiment, the created IWE was saved as a map and reused for more accurate motion estimation.

The estimated parameters are shown in Figure 11 along with the results of the homographic motion estimation and reference values. It can be seen that while the homographic motion estimation failed, the accuracy of our estimation was comparable to that of the motion estimation by the front RGB camera and light detection and ranging (LIDAR). Figure 12 shows the location obtained from the estimated θ reference value from ARKit4 and the plotting result of the created IWE map. The IWE map shows the carpeted surface without motion blur, indicating that the motion estimation was performed correctly even on a surface where robust feature point extraction and tracking were difficult. Because it is difficult for a normal camera to perform stable tracking from a surface as the one shown in the figure, this evaluation further demonstrated the effectiveness of our motion estimation method while using an event camera.

## 4. Discussion

Previous studies have shown that contrast maximization and entropy minimization can cause the loss function to have multiple extremes in some situations [33,34]. In this study, we clearly showed that when using contrast maximization and entropy minimization to perform homographic motion estimation for a plane, the loss function has multiple extremes and the motion estimation result deviates from the true value. Such errors are considered to be a serious problem, especially when event camera motion estimation is used for autonomous driving. To solve this problem, we proposed a method of bird’s-eye transformation of event data into two dimensions parallel to the road surface. As shown in the results of the loss landscape experiment, the proposed method mitigates the problem that the loss function becomes non-convex around the true value. As mentioned in the results of the quantitative experiments using the CARLA simulator, in the task of motion estimation with an event camera mounted at an angle, the proposed method outperforms the conventional method in terms of accuracy, number of computations, and computation time. Furthermore, in an experiment with an event camera attached to a dolly, we qualitatively showed that the proposed method is capable of motion estimation. From these experiments, we concluded that our method can accurately estimate the motion of an event camera mounted on a vehicle or robot by using contrast maximization, which has caused errors in the previous methods.

However, one limitation of our method is that although the computation time was reduced compared to the conventional method, real-time motion estimation is still not possible. As Gallego et al. stated, event camera algorithms can be broadly divided into two categories depending on how the events are processed: (1) those that operate on an event-by-event basis, where the arrival of a single event changes the state of the system, and (2) those that operate on a group of events [7]. Since we believe that the former method takes less time to estimate than the latter, we plan to improve our approach by modifying the contrast maximization method from processing groups of events to processing each event individually.

## 5. Conclusions

In this paper, we proposed a method of camera motion estimation using a road surface observed at an angle by an event camera. By using a bird’s-eye view transformation, events were transformed into two dimensions parallel with the road surface, which mitigates the problem of the loss function becoming a non-convex function, which occurs in conventional methods. As a result, the loss function becomes convex around the true value, which enables a wider range of the initial value to be taken, resulting in highly accurate and fast motion estimation. Experiments using CARLA and real data demonstrated the effectiveness of the proposed method in terms of both speed and accuracy.

## Figures and Tables

**Figure 1 sensors-22-00773-f001:**
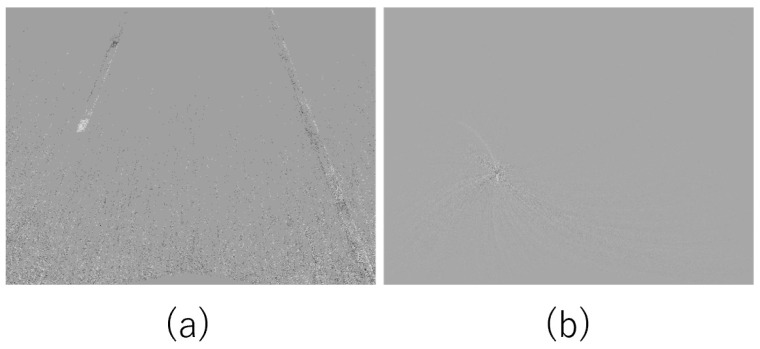
An example of failed estimation. (**a**) Events before estimation. (**b**) Events warped by the wrong estimation. The events have been warped onto the same point.

**Figure 2 sensors-22-00773-f002:**
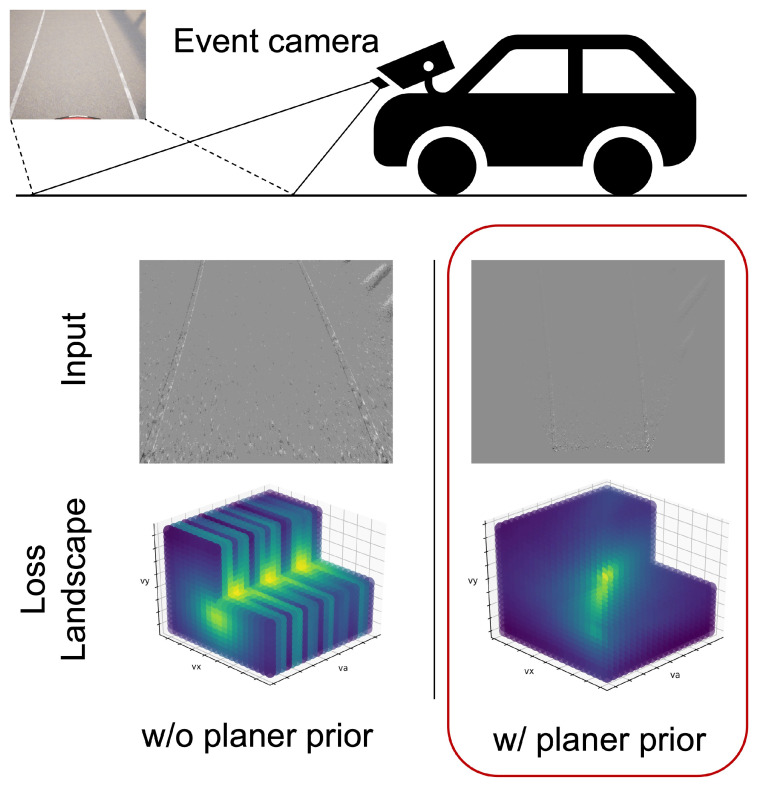
Overview of our method. When capturing a road surface at an angle for motion estimation, it can be difficult to track using an RGB camera. Even with an event camera, homographic motion estimation can have multiple extrema in the loss function. By performing a bird’s-eye view transformation, the loss function becomes convex around the true value, which allows for improvements in accuracy and speed.

**Figure 3 sensors-22-00773-f003:**
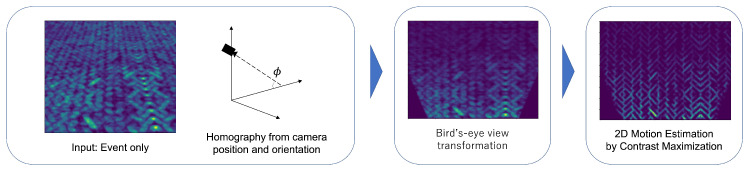
System flow. The input is only events. The events are transformed into a bird’s-eye view using the homography computed from the camera position and angle. The 2D motion estimation is then performed using contrast maximization.

**Figure 4 sensors-22-00773-f004:**
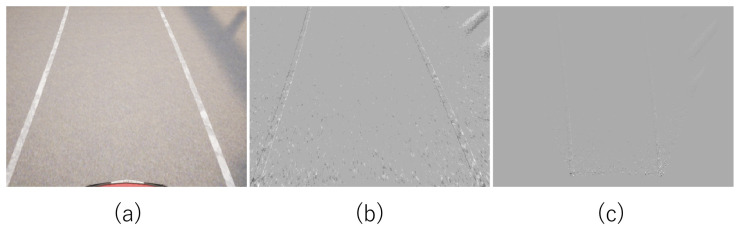
Road images taken at an angle as created by CARLA. (**a**) RGB image (not used). (**b**) Accumulated events. (**c**) Bird’s-eye view transformation of events.

**Figure 5 sensors-22-00773-f005:**
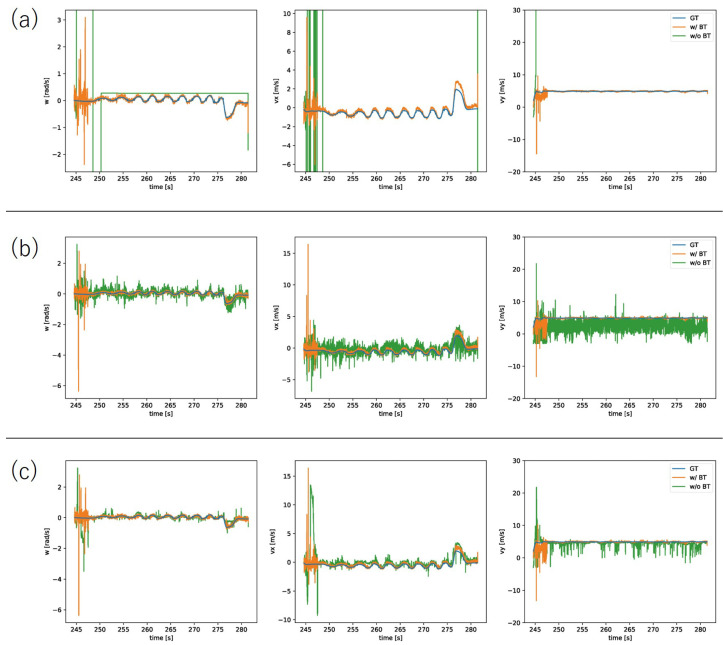
Results of the quantitative experiments. Blue: ground truth. Orange: proposed method with the bird’s-eye view transformation (w/BT). Green: entropy minimization method without the bird’s-eye view transformation (w/o BT). (**a**) Condition (a). (**b**) Condition (b). (**c**) Condition (c).

**Figure 6 sensors-22-00773-f006:**
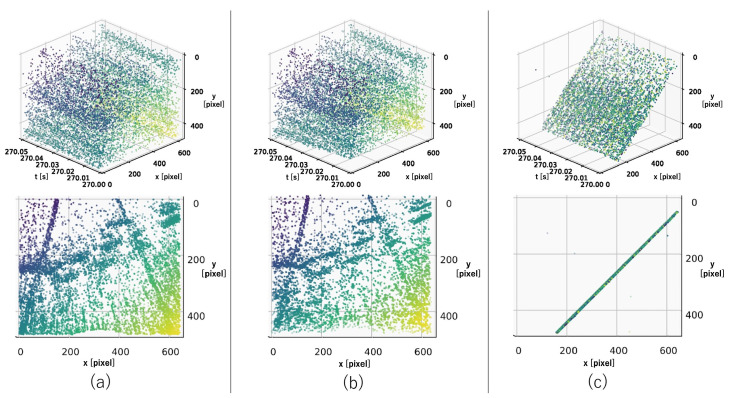
(**a**) Events before warp. (**b**) Events warped by the true value. (**c**) Events warped by a failed homographic motion estimation. The top row shows how the events were warped by homographic motion estimation in the XYT space. The bottom row shows how they are projected to the IWE.

**Figure 7 sensors-22-00773-f007:**
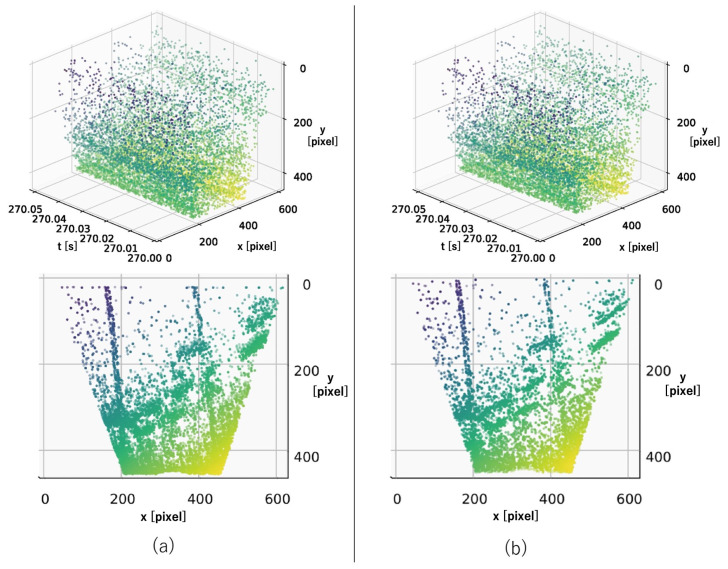
(**a**) Events before the warp. (**b**) Events warped by the isometric motion estimation. The top row shows how the events were warped by the isometric motion estimation in the XYT space. The bottom row shows how they were projected to the IWE.

**Figure 8 sensors-22-00773-f008:**
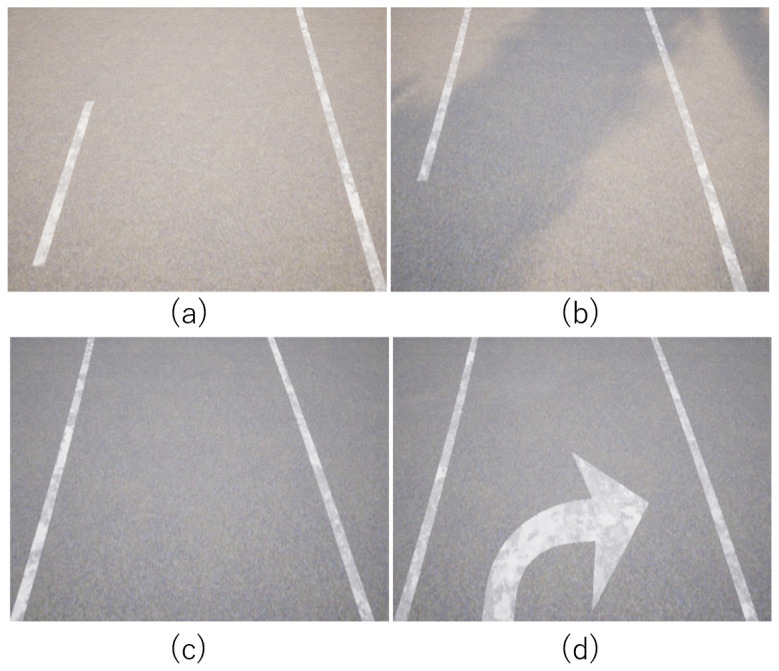
RGB images of scenes for the computational complexity experiments. (**a**) Scene 1. (**b**) Scene 2. (**c**) Scene 3. (**d**) Scene 4.

**Figure 9 sensors-22-00773-f009:**
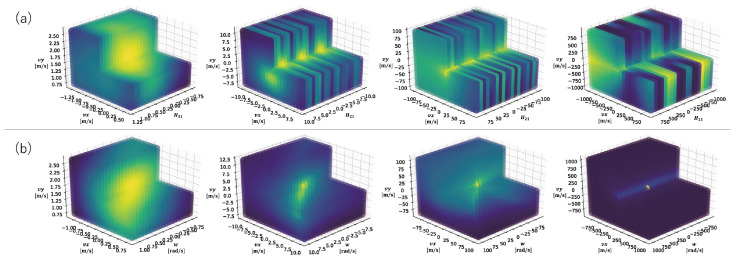
The landscape of the entropy near the true value, with and without the bird’s-eye view transformation. (**a**) Homographic motion estimation. $H_{21}$ refers to the (2,1) entry of the estimated homography matrix. (**b**) Proposed method.

**Figure 10 sensors-22-00773-f010:**
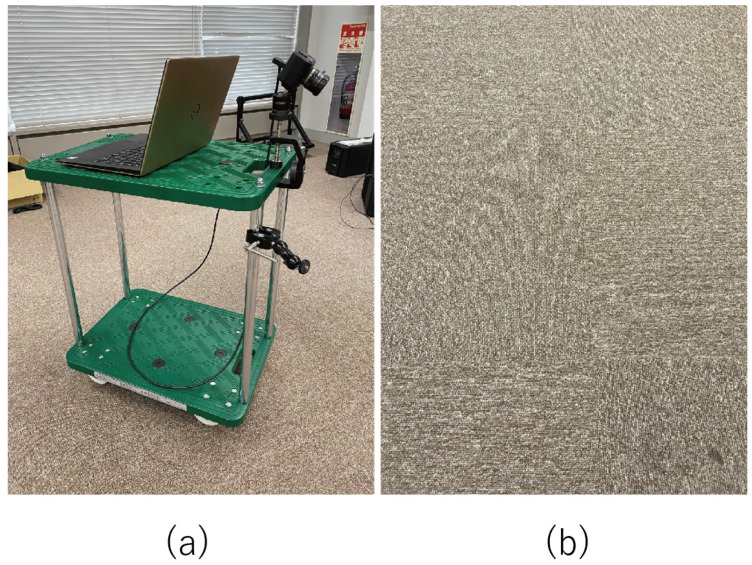
(**a**) Experimental setup. An event camera was mounted on a dolly at an angle (not parallel) to the surface of interest. (**b**) Carpeted surface; it would be difficult to extract its features with a normal camera.

**Figure 11 sensors-22-00773-f011:**
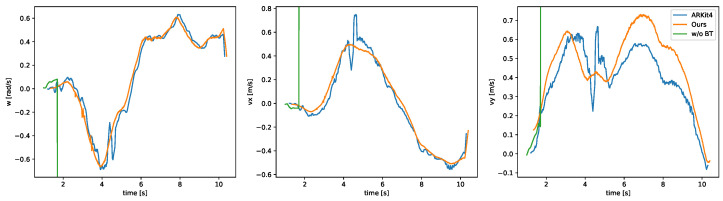
Results of the motion estimation experiment on a carpeted surface. Blue: ARKit4. Orange: proposed method with the bird’s-eye view transformation. Green: entropy minimization method without the bird’s-eye view transformation.

**Figure 12 sensors-22-00773-f012:**
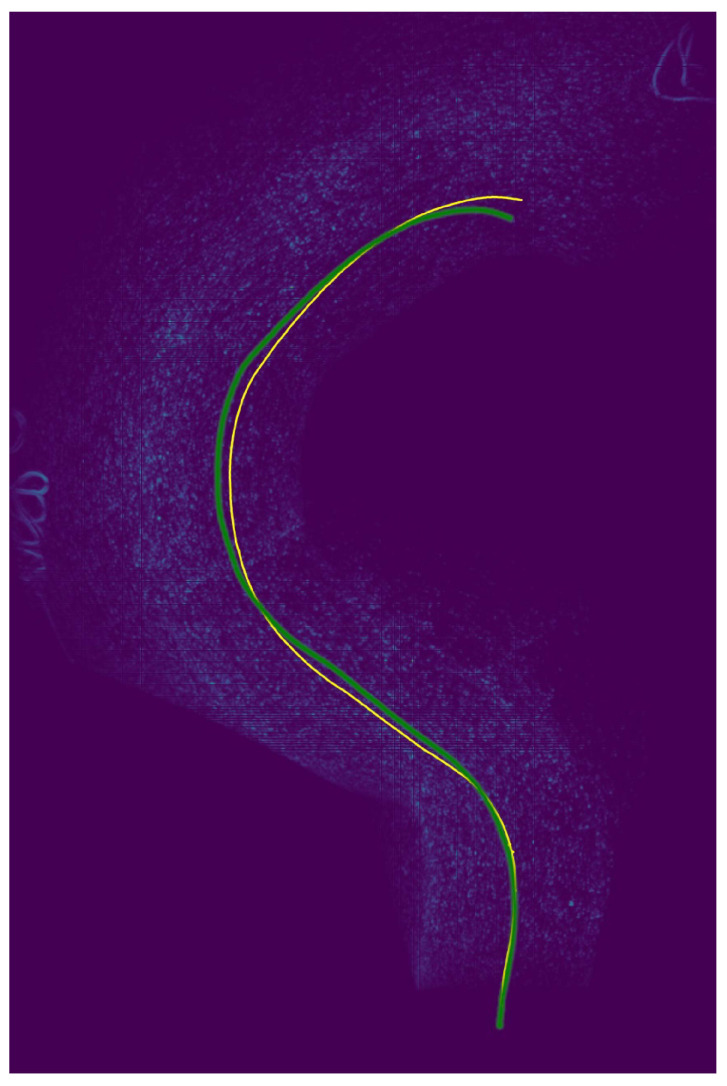
Result of our event camera motion estimation method on a carpeted surface (Green: location of the dolly obtained by ARKit4. Yellow: location of the dolly estimated by the proposed method). Our method was able to estimate motion even on a surface where robust feature point extraction and tracking were difficult.

**Table 1 sensors-22-00773-t001:** Iteration and total time for the motion estimation. The column “w/o BT” represents the result of homographic motion estimation without bird’s-eye view transformation, and the column “ours” represents the estimation result by the proposed method. The top performance is highlighted in bold.

Metric	Scenes	w/o BT	Ours
Iteration	Scene 1	200	**19**
	Scene 2	124	**19**
	Scene 3	118	**11**
	Scene 4	107	**29**
	Ave.	137.2	**19.5**
Total time (s)	Scene 1	3.98	**0.57**
	Scene 2	12.39	**3.98**
	Scene 3	15.94	**3.41**
	Scene 4	21.63	**8.61**
	Ave.	13.49	**4.14**

## Data Availability

Data sharing not applicable.

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
