# Peer review of "Accuracy and Speed Improvement of Event Camera Motion Estimation Using a Bird’s-Eye View Transformation"

_sensors, 2022, doi:10.3390/s22030773_

Round 1

Reviewer 1 Report

The authors proposed a method for motion estimation by optimizing contrast in bird’s-eye view space. The topic is interesting, but I have some concerns which should be resolved for acceptance.

  1. The authors should mention more related works.
  2. Comparison with state-of-the-art approaches is required.
  3. What are the advantages when your method is used?
  4. Which system did you use for Table 1? CPU? RAM?
  5. How did you visualize the loss landscape of Figure 9?

Reviewer 2 Report

The manuscript is well written and describes the advantages of motion estimation of event camera using a bird-eye view transformation. However, the current manuscript requires revisions before it can be accepted for publication. 

The main comments are given as follows: 

1) In this manuscript including the title, the authors repeatedly used "camera motion estimation" and "vehicle motion estimation". If the reviewer understands it correctly, these two terms refer to the same stuff because the camera is installed stationary to the vehicle and the minor relative motions are ignored. To avoid the confusion, the reviewer recommends the authors to revise them to a consistent term throughout the manuscript. 

2) In Section 1.1 and 1.2 of the introduction, the authors separately described the optical flow estimation and planar motion estimation. The reviewer might know the reason why the authors organized these two sub sections in the Introduction part. But the overall messages that the authors wanted to convey are not clear.

In this way, the reviewer recommends the author to revise the sub-section organization within Introduction following the flow: Backgrounds -- Motivations -- Objectives -- Related works -- Contributions. Also, there is no need to number the sub sections in the introduction, where the connections of the flow should be embedded into the writeup. 

3) As the authors emphasized in L172 to L174 of this manuscript (and some other lines throughout this manuscript), the novelty of this method lies in making loss function convex and reducing the iterations. The novelty sounds very interesting to the reviewer. However, the authors just empirically mentioned this novelty for several times throughout this manuscript without provide any mathematical analysis about the reason why the loss function is convex and why the optimization converges faster than than the one without the transformation. The reviewer understands the authors have provided extensive experimental results to demonstrate the advantages of applying the transformation, but they are more like an empirical experience instead of a derivation.

The reviewer recommends the authors to derive it mathematically because they are the most important contribution in this manuscript. Otherwise, the lack of mathematical analysis should be mentioned, and the novelty should be revised to be based on an "empirical" manner. 

Other comments:

1) In the legend of Figure 5, is BT equal to Bird's-eye view Transformation and GT equals to Ground Truth? Please clarify the abbreviations before the first time they appear to this manuscript.

2) In Table 1, is the column under Homography refering to the results without using Bird's-eye view transformation, e.g. w/o BT? Please clarify. 

Round 2

Reviewer 1 Report

My concerns are resolved.

Reviewer 2 Report

The reviewer thanks the authors to make such revisions and the current manuscript looks better. 

As for the point 3, although the authors did not provide mathematical analysis about the findings, these empirical findings still make sense and have novelty to the reviewer. 

In this way, the reviewer recommends the revised manuscript to be accepted for publication on Sensors.